# A New Sliding Mode Control Algorithm of IGC System for Intercepting Great Maneuvering Target Based on EDO

**DOI:** 10.3390/s22197618

**Published:** 2022-10-08

**Authors:** Kang Niu, Xi Chen, Di Yang, Jiaxun Li, Jianqiao Yu

**Affiliations:** School of Astronautics, Beijing Institute of Technology, Beijing 100081, China

**Keywords:** IGC system, sliding mode control, great maneuvering target, control algorithm, extended disturbance observer (EDO)

## Abstract

To intercept the great maneuvering target, combining with the sliding mode and the extended disturbance observer, a new control algorithm for integrated guidance and control (IGC) system is proposed in this paper. Firstly, the paper formulates the Missile–Target problem. Then the paper establishes an uncertain IGC dynamic model where the nonlinearities, the perturbations and the maneuvering of the target are regarded as disturbance. Secondly, a second-order disturbance observer is designed to estimate the disturbance and their derivatives.. After this, combining with the second-order disturbance observer, a modified sliding surface and the corresponding reaching law are designed to obtain the rudder deflection command directly. Thus, the real sense of IGC system is achieved. Next, the paper uses the Lyapunov stability theory to prove the stability of the system. Finally, the paper provides different simulation cases, which have different maneuver modes of the target, to demonstrate the superiority of the proposed method in reducing the response time, increasing the rudder response, and having a high interception probability.

## 1. Introduction

Nowadays, high precision guidance weapons have been widely used in modern military affairs, especially in the Gulf War (1991), the Afghan War (2001), and the Russia—Ukraine War (2022). Thus, a growing number of countries have been researching high performance, high speed and high precision weapon systems. Additionally, the integrated guidance and control (IGC) system has attracted researchers’ attention, due to much consideration about the rapid response, the least time delay and the coupling relation between the control and the guide.

As is well known, when designing a new guidance and control system for a missile, the traditional process usually neglects the coupling effect between the guidance loop and the control loop. These two subsystems are often designed separately, which is feasible when the target has little or no maneuver capability. However, for high speed and great maneuver targets, the relative movement between the missile and the target changes rapidly and the coupling effect will become stronger. Furthermore, the bandwidth of the guidance loop will restrict the interception procedure. In addition, using the traditional design methods, the performance of the missile may degrade and the closed loop of the missile may even be instable.

Compared with the traditional design process, the IGC system can calculate the rudder deflection command directly from the relative motion information between missile and target. This method can fully consider the coupling effect between the guidance loop and the control, and avoid the time delay between these two loops. Furthermore, the procedure of the IGC can also reduce the iterative process. Thus, it is extremely advantageous to intercept the maneuvering targets. Therefore, many studies had been investigated on the IGC controller design. Considering the field-of-view (FOV) constraint, Zhao [1] developed a new IGC control method for missiles with a strapdown seeker by using a neural network. However, the speed of convergence is slow and its interception accuracy is not good. Shima [2] developed an IGC scheme by using the SMC technique. However, the chattering problem of SMC had not been resolved, and the rapidity of the IGC system also needs to be improved. To improve the performance of sensors which have look-angle requirements, Shaferman [3] proposed a new control law that enables controlling the look-angle based on linear missile dynamics in a linear optimal control framework. However, it requires the time-to-go information which is usually a formidable challenge. Menon [4] designed a new control method for the integrated guidance and autopilot system by using the feedback linearization technique. However, the coupling effect between the guidance and the control loop is ignored. To improve the interception performance, Wang [5] developed a novel control strategy for the integrated guidance and control design subject to input constraints. In respect of an unknown maneuvering target with multiple uncertainties and control constraint, Shao [6] proposed a novel composite integrated guidance and control (IGC) law by using the back-stepping technique and the extended state observer (ESO). While the paper also separated the IGC system into the guidance loop and the control loop. Considering the type of the target maneuvers and the missile model uncertainties, Yan [7] proposed a new control law for the IGC system by using the small-gain theorem. However, the interception performance was not better, especially with regards to intercepting high maneuvering targets. Kim [8] transformed the IGC control problem into a single finite-horizon optimization problem with various terminal constraints and it will be explicitly solved in the discrete-time domain for terminal homing. Regarding the maneuvering target, Park [9] proposed a novel control law for the IGC system by using model predictive control (MPC) technology. However, the accuracy of the model has a great influence on the performance of MPC. To intercept and rendezvous the maneuvering target within an obstacle’s environments, Weiss [10] designed a new control algorithm by using linear quadratic optimal control theory. Ming [11] proposed a new control method for the IGC system based on approximations of the Hamilton—Jacobi—Bellman equation. However, the disturbances and uncertainties of the 6-DOF missile model were not considered. When considering the constrained impact angle, a novel IGC control algorithm was investigated with a non-singular terminal sliding mode control (NTSMC) technique in reference [12]. However, the fatal disadvantage is that this approach cannot obtain the rudder deflection command directly. Therefore, it is not a real type of IGC system. Wang [13] proposed a new IGC control law for dual-control missiles based on the backstepping sliding mode design approach. However, the computation explosion problem of the conventional backstepping design method was ignored. Liu [14] developed a new robust 3D-IGC control approach for the STT hypersonic missile with high uncertainties and strict impact angle constraints by using the dynamic inversion theory. Hou [15] proposed a novel integrated guidance and autopilot design method for homing missiles based on the adaptive block dynamic surface control approach. However, the interception performance is not superior when up against large maneuvering targets. Xiong [16] uses the dynamic surface control and extended state observer techniques to establish a new three-dimensional IGC control law for a BTT missile attacking a ground fixed target in the presence of input saturation and actuator failure. However, it is clear that influences from the change of velocity in terminal flight are always ignored. Zhou [17] designed a new three-dimensional IGC control law based on the fractional integral terminal sliding mode control (FITSMC) scheme for STT missiles with an impact angle constraint. However, the system’s uncertainties and unknown disturbances have not been resolved. Su [18] proposed a multivariable NTSMC strategy for six-degree-of-freedom reusable launch vehicles by using the disturbance observer technique. Su [19] proposed a new robust adaptive backstepping control (RABC) strategy to solve the problem caused by the large range of flight envelopes for reusable launch vehicles. However, this strategy consists of two-loop controllers designed via the backstepping method. The outer loop can track the desired attitude by the design of virtual control the desired angular velocity, and the inner one can track the desired angular velocity by the design of the control torque. Wang [20] proposed a new adaptive IGC control law for hypersonic vehicles in dive phase, when considering the 3D impact angles, the commands of Euler angles, the normal overload and the three-channel body rates. However, it is obvious that this approach did not consider the FOV constraints. Wang [21] established a new integrated guidance and control law by using the dynamic surface control theory, when considering multiple missiles attacking targets cooperatively. Tee [22] investigated a new control algorithm for nonlinear systems with state constraints by using Integral Barrier Lyapunov Functionals (iBLF). Although, the core idea of this method is still to use the backstepping control strategy. In recent years, with the rapid development of machine learning theory, a neural network has been applied to the IGC system. Yu [23] developed a novel fixed-time fast nonsingular terminal slide mode surface (FNTSMS) to addresses the simultaneous attack of multiple missiles against a maneuvering target in 3D space. By using this method, all missiles can attack the target at the same desired time. Zhou [24] developed a novel IGC control algorithm by combining the neural network with the backstepping control theory. However, the convergence rate is relatively slow when estimating the uncertainty terms in the IGC model. In addition, Gaudet [25] proposed a novel IGC guidance law that uses observations consisting solely of seeker line-of-sight angle measurements and their rate of change. Although it introduces the reinforcement meta-learning neural network, which just participates in optimizing the new policy. 

In terms of the methods mentioned above, most of these often use the idea of BC, which did not achieve the true concept of IGC. Furthermore, using a neural network to get the rudder deflection command often takes a large number of iterations, which is not suitable for the real battlefield scene or for intercepting the great maneuvering target. In addition, the conventional SMC method often chose a linear sliding surface, which has the fatal shortcomings in that it cannot eliminate the chattering phenomenon, and the state tracking error will not converge to zero in a limited time. Therefore, to solve all these problems, a new sliding mode control algorithm of the IGC System has been designed. The major contributions in this paper are summarized as follows:(1)A new IGC nonlinear mathematic dynamic model, which contains the uncertain terms, is established. In this model, the nonlinearities, and the perturbations caused by variations of aerodynamic parameters are viewed as disturbance.(2)A new IGC method by using a modified sliding surface, which contains all state variables instead of backstepping technique, is proposed to obtain control command directly. Thus, it can achieve the true concept of an IGC system.(3)To estimate the uncertainties and their derivatives, the paper designs the second-order disturbance observer to compensate for the proposed IGC controller.(4)The IGC system’s stability is strictly proved with the Lyapunov theory, and the contrastive simulation results are presented to verify the effectiveness and superiority of the designed method.

This paper is organized as follows: the introduction of IGC is stated in Section 1. Then the IGC dynamic model is formulated in Section 2. The design of the IGC law and its stability analysis are described in Section 3. Simulation performance is illustrated in Section 4. Finally, the conclusion is presented in Section 5.

## 2. Problem Formulation

Taking the pitch plane as an example, the paper describes the relative motion of the Missile–Target and the pursuit of it is shown in Figure 1. The dynamic schematic of the interceptor steered by the tail aerodynamic control surfaces is shown in Figure 2.

Figure 1 denotes the pursuit geometry of the missile–target in the pitch plane. XOY is the inertial coordinate system. $v_{m},\;\theta_{m}$ and $a_{m}$ represent the velocity, flight path angle and normal acceleration of the missile, respectively. $v_{T},\;\theta_{T}$ and $a_{T}$ denote the velocity, flight path angle and normal acceleration of the target, respectively. R is the relative distance between the missile and the target, and q is the LOS angle. In addition, $x_{m},\;y_{m}$ and $x_{T},\;y_{T}$ denote the current position of the missile and target in the inertial coordinate system, respectively. $O_{m}x_{b}$ is body axis of the missile, and ϑ is the pitch angle of the missile.

Figure 2 depicts a schematic of the missile steered by the effects of tail aerodynamic control surfaces in the pitch plane. $M_{g},\;X_{aero}$ and $Y_{aero}$ denote the gravity, the aerodynamic drag and the lift force of the missile, respectively. In addition, α is the attack angle.

### 2.1. Subsection Missile-Target Engagement Kinematics

According to Figure 1 and Figure 2, assuming that $v_{m}=0,\;v_{T}=0$, the missile–target engagement dynamics are formulated as follows.
(1)$$\dot{R}=v_{T}\cos\left(\theta_{T}-q\right)-v_{m}\cos\left(\theta_{m}-q\right)$$
(2)$$R\dot{q}=v_{T}\sin\left(\theta_{T}-q\right)-v_{m}\sin\left(\theta_{m}-q\right)$$
(3)$${\dot{\theta }}_{T}=\frac{a_{T}}{v_{T}}$$
(4)$${\dot{\theta }}_{m}=\frac{a_{m}}{v_{m}}$$

Then, differentiating (1) and combining with (2) yields
(5)$$R\ddot{q}+2\dot{R}\dot{q}=a_{T}\cos\left(\theta_{T}-q\right)-a_{m}\cos\left(\theta_{m}-q\right)$$

### 2.2. Missile Dynamics

In this paper, the nonlinear dynamic model for the missile in the pitch plane is as follows
(6)$$\left\{ \begin{array}{l} a_{m}=n_{y}g=\frac{Y}{m}-g\cos\theta_{m}+\omega_{z}\\ \dot{\vartheta }=\omega_{z}\\ J_{z}{\dot{\omega }}_{z}=M_{0}\left(\alpha ,M_{a},h,V_{m},\omega_{z}\right)+M_{z}\delta_{z}\\ \alpha =\vartheta -\theta_{m} \end{array}\right.$$
where Y is the lift force. $n_{y}$ is the normal overload. g is the acceleration of gravity. ϑ is the pitch angle. $\omega_{z}$ is the pitch rate. α is the attack angle. $J_{z}$ is the moment of inertia about z-axis. $\delta_{z}$ is the deflection angle for pitch control. $M_{\delta_{z}}$ represents the control contribution to the angular acceleration. $M_{0}$ represents the angular acceleration contributions from all other sources and it is approximated as $M_{0}=M_{\alpha }\alpha +M_{\omega z}\omega_{z}$ here $M_{\alpha }$ is the angular acceleration contributions from the attack angle, and $M_{\omega z}$ is the angular acceleration contribution to the pitch rate.

The lift force and relative parameters are defined as follows.
(7)$$\left\{ \begin{array}{l} Y=57.3QSc_{y}^{\alpha }\alpha \\ M_{\alpha }=57.3QSLm_{z}^{\alpha }\\ M_{\omega_{z}}=\frac{QSL^{2}m_{z}^{\omega_{z}}}{V_{m}}\\ M_{\delta_{z}}=57.3QSLm_{z}^{\delta_{z}} \end{array}\right.$$
where Q is the dynamic pressure, which is defined as $Q=0.5\rho V_{m}^{2}$. S is the aerodynamic reference area. L is the reference length. $c_{y}^{\alpha }$ is the lift force derivative with respect to α. $c_{y}^{\delta_{z}}$ is the lift force derivative with respect to $\delta_{z}$. $M_{\alpha },\;M_{\omega z}$ and $M_{\delta_{z}}$ denote the moment determined by aerodynamic parameters such as the angle of attack α, pitch angular rate $\omega_{z}$, and deflection angle for pitch control $\delta_{z}$.

According to Equations (3)–(6), when differentiating $n_{y}$, we obtain the following equation.
(8)$${\dot{n}}_{y}=\frac{57.3QSc_{y}^{\alpha }}{mg}\omega_{z}+\left(\frac{g\sin\left(\theta_{m}\right)}{v_{m}}-\frac{57.3QSc_{y}^{\alpha }}{mv_{m}}\right)n_{y}+d_{n_{y}}$$

According to Equations (3)–(8), the missile dynamic model is as follows.
(9)$$\left\{ \begin{array}{l} a_{m}=n_{y}g=\frac{57.3QSc_{y}^{\alpha }\alpha }{m}-g\cos\theta_{m}+\omega_{z}\\ \dot{\vartheta }=\omega_{z}\\ J_{z}{\dot{\omega }}_{z}=57.3QSLm_{z}^{\alpha }\alpha +\frac{QSL^{2}m_{z}^{\omega_{z}}}{V_{m}}\omega_{z}+57.3QSLm_{z}^{\delta_{z}}\delta_{z}\\ \alpha =\vartheta -\theta_{m}\\ {\dot{n}}_{y}=\frac{57.3QSc_{y}^{\alpha }}{mg}\omega_{z}+\left(\frac{g\sin\left(\theta_{m}\right)}{v_{m}}-\frac{57.3QSc_{y}^{\alpha }}{mv_{m}}\right)n_{y}+d_{n_{y}} \end{array}\right.$$

### 2.3. The Uncertain Linear IGC Model

In this section, to establish the uncertain linear system, the nonlinearities, the perturbations caused by variations of aerodynamic parameters and other terms are viewed as disturbance. Before constructing the uncertain linear IGC model, we will first define $V_{rq}=R\dot{q}$. Then differentiating $V_{rq}$, together with Equation (5) yields
(10)$${\dot{V}}_{rq}=-\frac{\dot{R}}{R}V_{rq}+a_{T}\cos\left(\theta_{T}-q\right)-a_{m}\cos\left(\theta_{m}-q\right)$$

Then, we can define $x_{1}=V_{rq}/\left(-Y/\alpha m\right)$, $x_{2}=\alpha$, $x_{3}=\omega_{z}$ and $u=\delta_{z}$. Thus, the state vector is defined as $X=\left[x_{1},x_{2},x_{3}\right]$. Then, considering the model error and regarding the nonlinearities, the perturbations caused by variations of aerodynamic parameters and other terms as disturbance, the IGC dynamic model with the strict-feedback state equation can be formulated as follows.
(11)$$\left\{ \begin{array}{l} {\dot{x}}_{1}=x_{2}+d_{1}^{*}\\ {\dot{x}}_{2}=x_{3}+\kappa_{1}x_{2}+d_{2}^{*}\\ {\dot{x}}_{3}=bu+\kappa_{2}x_{3}+\kappa_{3}x_{2}+d_{3}^{*} \end{array}\right.$$
where $d_{1}^{*},d_{2}^{*},d_{3}^{*}$ are defined as follows.
(12)$$\begin{array}{l} d_{1}^{*}=\left({\dot{\theta }}_{m}-\frac{\dot{R}}{R}\right)x_{1}+\frac{a_{T}\cos\left(\theta_{T}-q\right)}{-57.3QSc_{y}^{\alpha }/m}\\ \;+\frac{g\cos\theta_{m}\cos\left(\theta_{m}-q\right)}{-57.3QSc_{y}^{\alpha }/m}+{\tilde{d}}_{1}\\ d_{2}^{*}=\kappa_{4}+d\left(c_{y}^{\alpha }\right)+{\tilde{d}}_{2}\\ d_{3}^{*}=d\left(m_{z}^{\alpha },m_{z}^{\omega_{z}},m_{z}^{\delta_{z}}\right)+{\tilde{d}}_{3} \end{array}$$

Equations (11) and (12) show ${\tilde{d}}_{1},\;{\tilde{d}}_{2}$ and ${\tilde{d}}_{3}$ denote the model error. The disturbance $d_{1}^{*}$ contains the unknown target acceleration, time-varying nonlinearities with respect to system states and the model error. $d_{2}^{*}$ contains the time-varying perturbations caused by variations of aerodynamic parameters and $d\left(c_{y}^{\alpha }\right)$, $d_{3}^{*}$ $d\left(m_{z}^{\alpha },m_{z}^{\omega_{z}},m_{z}^{\delta_{z}}\right)$ and the model error.

As in Equation (11), $\kappa_{1},\;\kappa_{2},\;\kappa_{3},\;\kappa_{4}$ and b are defined as follows
(13)$$\left\{ \begin{array}{l} \kappa_{1}=\frac{1}{mV_{m}}57.3QSc_{y}^{\alpha },\kappa_{2}=\frac{QSL^{2}m_{z}^{\omega_{z}}}{J_{z}V_{m}}\\ \kappa_{3}=\frac{57.3QSLm_{z}^{\alpha }}{J_{z}},\kappa_{4}=\frac{g\cos\theta_{m}}{V_{m}}\\ b=\frac{57.3QSLm_{z}^{\delta_{z}}}{J_{z}} \end{array}\right.$$

Thus, the IGC dynamic model becomes a linear 3rd order single input system affected by uncertain disturbances.

## 3. IGC Controller Design

As is well known, the conventional SMC method often chose a linear sliding surface, which has the fatal shortcomings that it cannot eliminate the chattering phenomenon and the state tracking error will not converge to zero in a limited time. To solve all these problems, a new sliding mode control algorithm of the IGC system was designed. In this section, the paper will present a modified sliding surface to obtain the control law directly and it provides a design of a second-order disturbance observer to estimate the disturbances and their derivatives. Before establishing the observer, it is necessary to provide some assumptions considered in this paper.

**Assumption** **1.**
*The disturbances*

$d_{i}^{*},\;\left(i=1,2,3\right)$

*are continuous, and satisfies the following condition.*

(14)
$$\left|\frac{d^{j}d_{i}^{*}\left(t\right)}{dt^{j}}\right|\leq \mu ,\;\left(i=1,2,3,j=0,1,2\right)$$

*where*

μ

*is a positive number.*


### 3.1. A New Modified Sliding Surface and the Controller Design

As mentioned in the introduction section, many existing control algorithms are designed for the IGC system. All of these methods mostly used the traditional separation idea that the guidance command is given by the guidance loop first, and then the missile is controlled by the control loop to achieve the guidance command, which is not the true concept of IGC. Furthermore, there is the fatal shortcoming of time delay, which is caused by the command filter.

Firstly, according to Equation (11), a conventional sliding surface is given as follows
(15)$$\bar{s}=c_{1}x_{1}+c_{2}x_{2}+x_{3}$$
where
$c_{1}$ and $c_{2}$ are positive constants.

**Remark** **1.***In this paper, the purpose of missile control is that the angular rate is equal to zero*$\dot{q}=0$*. The initial form of sliding mode surface will be governed by*(16)$$\begin{array}{l} \tilde{s}=c_{1}x_{1}+c_{2}{\dot{x}}_{1}+{\ddot{x}}_{1}\\ =c_{1}x_{1}+\left(c_{2}+\kappa_{1}\right)x_{2}+x_{3}+c_{2}d_{1}^{*}+d_{2}^{*}+{\dot{d}}_{1}^{*} \end{array}$$*where*$c_{1}>0,c_{2}>0$*, and the larger*$c_{2}$*is, the faster the sliding mode converges*.

A mentioned in previous section, the dynamic model error, the nonlinearities, target maneuvers, perturbations, etc. are viewed as disturbance in this paper. Thus, considering all those state variables and all estimates of the unmatched disturbances as well as their derivatives. And combining with Equations (15) and (16), it is obviously that the disturbances are unknown. Thus, they can be replaced by the estimates. In addition, to further improve the speed of convergence, the nonlinear exponential function is introduced to the sliding mode surface. Thus, the novel sliding mode surface can be modified as follows.
(17)$$\left\{ \begin{array}{l} \hat{s}=c_{1}x_{1}+\left(c_{2}+\kappa_{1}\right)x_{2}+x_{3}+c_{2}{\hat{d}}_{1}^{*}+{\hat{\dot{d}}}_{1}^{*}+{\hat{d}}_{2}^{*}\\ s=\hat{s}-\hat{s}\left(0\right)e^{-\gamma t} \end{array}\right.$$
where γ is a positive constant. ${\hat{d}}_{1}^{*},{\hat{d}}_{2}^{*}$ and ${\hat{\dot{d}}}_{1}^{*}$ are the estimate of $d_{1}^{*},d_{2}^{*}$ and ${\dot{d}}_{1}^{*}$ respectively.

Differentiating Equation (17), together with Equation (11), the reaching law and the control input are designed as follows
(18)$$\begin{array}{l} \dot{s}=-\left(k_{l}s+k_{s}\mathrm{sgn}\left(s\right)\right)+c_{1}{\tilde{d}}_{1}^{*}+\\ \;\left(c_{2}+\kappa_{1}\right){\tilde{d}}_{2}^{*}+{\tilde{d}}_{3}^{*}+c_{2}{\dot{\hat{d}}}_{1}^{*}+{\dot{\hat{d}}}_{2}^{*}+{\hat{\dot{d}}}_{1}^{*} \end{array}$$
(19)$$u=-\frac{1}{b}\left(\begin{array}{l} \left(c_{1}+c_{2}\kappa_{1}+\kappa_{1}\kappa_{1}+\kappa_{3}\right)x_{2}+\left(c_{1}+\kappa_{1}+\kappa_{2}\right)x_{3}\\ +\left(c_{2}+\kappa_{1}\right){\hat{d}}_{2}^{*}+c_{1}{\hat{d}}_{1}^{*}+{\hat{d}}_{3}^{*}+\gamma \overset{⌢}{s}\left(0\right)e^{-\gamma t}\\ +k_{l}s+k_{s}\mathrm{sgn}\left(s\right) \end{array}\right)$$
where $k_{l}>0,k_{s}>0$, and the term $-k_{l}s/b$ in (18) helps alleviate chatter by making it possible to use a smaller $k_{s}$ and assure ultimate boundedness of s.

**Remark** **2.**
*To reduce the chattering phenomenon, the sign function is replaced by the saturation function in this paper. Thus, the sliding mode controller can be written as follows.*

(20)
$$u=-\frac{1}{b}\left(\begin{array}{l} \left(c_{1}+c_{2}\kappa_{1}+\kappa_{1}\kappa_{1}+\kappa_{3}\right)x_{2}+\left(c_{1}+\kappa_{1}+\kappa_{2}\right)x_{3}+\\ \left(c_{2}+\kappa_{1}\right){\hat{d}}_{2}^{*}+c_{1}{\hat{d}}_{1}^{*}+{\hat{d}}_{3}^{*}+\gamma \overset{⌢}{s}\left(0\right)e^{-\gamma t}\\ +k_{l}s+k_{s}\mathrm{sat}\left(s\right) \end{array}\right)$$



### 3.2. Stability Analysis

In this section, the paper will use Extension of Disturbance Observer to estimate the disturbances and their derivatives firstly. Then the paper will give the stability analysis of the controller by using Lyapunov stability theory. 

Firstly, the disturbance $d_{i}^{*}\left(i=1,2,3\right)$ and its derivatives can be estimated by
(21)$$\left\{ \begin{array}{l} {\hat{d}}_{i}^{*}=p_{i1}+l_{i1}x_{i}\\ {\dot{p}}_{i1}=-l_{i1}\left(x_{i+1}+{\hat{d}}_{i}^{*}\right)+{\hat{\dot{d}}}_{i}^{*}\\ {\hat{\dot{d}}}_{i}^{*}=p_{ir}+l_{ir}x_{i}\\ {\dot{p}}_{ir}=-l_{ir}\left(x_{i+1}+{\hat{d}}_{i}^{*}\right) \end{array}\right.$$

In Equation (21), ${\hat{d}}_{i}$ and ${\hat{\dot{d}}}_{i}$ denote the error in the estimation of $d_{i}^{*}$ and ${\dot{d}}_{i}^{*}$ respectively, $p_{i1},p_{ir}$ the auxiliary variables, and $l_{i1},l_{ir}$ are user chosen constants.

Then, let the estimation errors, ${\tilde{e}}_{i}$ be defined as
(22)$${\tilde{e}}_{i}={\left[{\tilde{d}}_{i}^{*}\;{\tilde{\dot{d}}}_{i}^{*}\cdots {\tilde{d}}_{i}^{*r}\right]}^{T}$$
where ${\tilde{d}}_{i}^{*},\;{\tilde{\dot{d}}}_{i}^{*},\;\cdots ,\;{\tilde{d}}_{i}^{*r}$ are defined as
(23)$$\left\{ \begin{array}{l} {\tilde{d}}_{i}^{*}=d_{i}^{*}-{\hat{d}}_{i}^{*}\\ {\tilde{\dot{d}}}_{i}^{*}={\dot{d}}_{i}^{*}-{\hat{\dot{d}}}_{i}^{*}\\ \vdots \\ {\tilde{d}}_{i}^{*r}=d_{i}^{*r}-{\hat{d}}_{i}^{*r} \end{array}\right.$$
where ${\tilde{d}}_{i}^{*}$ denotes the error in the estimation of, $d_{i}^{*}$ ${\tilde{\dot{d}}}_{i}^{*}$ denotes the error in the estimation of ${\dot{d}}_{i}^{*}$.

**Theorem** **1.**
*Based on the proposed controller (20) and the EDO (21) for the proposed IGC model (11) with mismatched uncertainties, if the estimation error, the reaching law are bounded, the reaching law is bounded.*


Firstly, according to Equations (11) and (21), we obtain the following equation.
(24)$$\begin{array}{l} {\dot{\hat{d}}}_{i}^{*}=l_{i1}{\tilde{d}}_{i}^{*}+{\hat{\dot{d}}}_{i}^{*} \end{array}$$

Differentiating Equation (23), together with Equation (24), then
(25)$$\begin{array}{l} {\dot{\tilde{d}}}_{i}^{*}={\dot{d}}_{i}^{*}-l_{i1}{\tilde{d}}_{i}^{*}-{\hat{\dot{d}}}_{i}^{*}\\ \;=-l_{i1}{\tilde{d}}_{i}^{*}+{\tilde{\dot{d}}}_{i}^{*} \end{array}$$

Differentiating ${\tilde{\dot{d}}}_{i}^{*}$ and together with
(26)$${\dot{\tilde{\dot{d}}}}_{i}^{*}=-l_{ir}{\tilde{d}}_{i}^{*}+{\ddot{d}}_{i}^{*}$$

Differentiating (25) and using (26)
(27)$${\ddot{\tilde{d}}}_{i}^{*}=-l_{i1}{\dot{\tilde{d}}}_{i}^{*}-l_{ir}{\tilde{d}}_{i}^{*}+{\ddot{d}}_{i}^{*}$$

According the per Assumption 1, it is necessary and sufficient to select $l_{i1}>0$ and $l_{ir}>0$ for stability of ${\tilde{d}}_{i}^{*}$. Thus, the observer error dynamics can be expressed in compact form as
(28)$${\dot{\tilde{e}}}_{i}=D_{i}{\tilde{e}}_{i}+E_{i}d_{i}^{*r}$$
where $D_{i}$ and $E_{i}$ are defined as follows
(29)$$D_{i}=\left[\begin{array}{ccccc} -l_{i1} & 1 & 0 & 0 & 0\\ -l_{i1} & 0 & 1 & 0 & 0\\ \vdots & 0 & 0 & 1 & 0\\ -l_{ir-1} & 0 & 0 & 0 & 1\\ -l_{ir} & 0 & 0 & 0 & 0 \end{array}\right],E_{i}=\left[\begin{array}{c} 0\\ 0\\ \vdots \\ 0\\ 1 \end{array}\right]$$

According to Equation (29), it is obvious that the gains $l_{ir}$ of the EDO for the disturbances $d_{i}^{*}$ can always be chosen so that the eigenvalues of each $D_{i}$ are in LHP. Therefore, it is always possible to find a positive definite matrix $P_{i}$ such that
(30)$$D_{i}^{T}P_{i}+P_{i}D_{i}=-Q_{i}$$

For any given positive definite matrix $Q_{i}$, let $\lambda_{mi}$ denote the smallest eigenvalue of *Q_i_*. Then we can define a Lyapunov function as follows
(31)$$V\left({\tilde{e}}_{1},{\tilde{e}}_{1},\cdots ,{\tilde{e}}_{n}\right)=\sum_{i=1}^{n}{\tilde{e}}_{i}^{T}P_{i}{\tilde{e}}_{i}$$

Differentiating $V\left({\tilde{e}}_{1},{\tilde{e}}_{2},\cdots ,{\tilde{e}}_{n}\right)$ and together with Equation (28), yields
(32)$$\begin{array}{ll} \dot{V}\left({\tilde{e}}_{1},{\tilde{e}}_{2},\cdots ,{\tilde{e}}_{n}\right) & =\sum_{i=1}^{n}{\tilde{e}}_{i}^{T}\left(D_{i}^{T}P_{i}+P_{i}D_{i}\right){\tilde{e}}_{i}+2\sum_{i=1}^{n}{\tilde{e}}_{i}^{T}\left(P_{i}E_{i}\right){\ddot{d}}_{i}^{*}\\ & \leq -\sum_{i=1}^{n}{\tilde{e}}_{i}^{T}Q_{i}{\tilde{e}}_{i}+2\sum_{i=1}^{n}\left\|P_{i}E_{i}\right\|\left\|{\tilde{e}}_{i}\right\|\mu_{i}\\ & \leq -\lambda_{mi}\sum_{i=1}^{n}{\left\|{\tilde{e}}_{i}\right\|}^{2}+2\sum_{i=1}^{n}\left\|P_{i}E_{i}\right\|\left\|{\tilde{e}}_{i}\right\|\mu_{i}\\ & \leq -\left\|{\tilde{e}}_{i}\right\|\left(\lambda_{mi}\sum_{i=1}^{n}\left\|{\tilde{e}}_{i}\right\|-2\sum_{i=1}^{n}\left\|P_{i}E_{i}\right\|\mu_{i}\right) \end{array}$$

According to Equation (32), after a sufficiently long time, the norm of the estimation error is bounded by
(33)$$\left\|{\tilde{e}}_{i}\right\|\leq \frac{2\left\|P_{i}E_{i}\right\|\mu_{i}}{\lambda_{mi}},\left(i=1,2,\cdots ,n\right)$$

Let λ^imax≤max[2‖PiEi‖μiλmi] for all i, then $\left\|{\tilde{e}}_{i}\right\|\leq {\hat{\lambda }}_{i}^{\max}$. Thus, the estimation error is bounded.

Next, we will demonstrate that the reaching law is bounded. Firstly, let define a Lyapunov function related the control surface as follows
(34)$$V\left(s,\dot{s}\right)=\frac{s^{2}}{2}$$

Differentiating $\dot{V}\left(s,\dot{s}\right)$ and together with Equations (11), (24) and (33), yields
(35)$$\begin{array}{ll} \dot{V}\left(s,\dot{s}\right) & =s\dot{s}\\ & =-k_{l}s^{2}-k_{s}\mathrm{sgn}\left(s\right)s+\left(c_{1}{\tilde{d}}_{1}^{*}+\left(c_{1}+\kappa_{1}\right){\tilde{d}}_{2}^{*}+{\tilde{d}}_{3}^{*}+c_{2}{\dot{\hat{d}}}_{1}^{*}+{\dot{\hat{d}}}_{2}^{*}+{\dot{\hat{\dot{d}}}}_{1}^{*}\right)s\\ & =-k_{l}s^{2}-k_{s}\left|s\right|+s\left(\begin{array}{l} c_{1}{\tilde{d}}_{1}^{*}+\left(c_{1}+\kappa_{1}\right){\tilde{d}}_{2}^{*}+{\tilde{d}}_{3}^{*}+c_{2}\left(l_{11}{\tilde{d}}_{1}^{*}+{\dot{d}}_{1}^{*}-{\tilde{\dot{d}}}_{1}^{*}\right)+\\ l_{21}{\tilde{d}}_{2}^{*}+{\dot{d}}_{2}^{*}-{\tilde{\dot{d}}}_{2}^{*}+l_{12}{\tilde{d}}_{1}^{*} \end{array}\right)\\ & \leq -k_{l}s^{2}-k_{s}\left|s\right|+s\left(\begin{array}{l} \left(c_{1}+c_{2}l_{11}-c_{2}+l_{12}\right)\left\|e_{1}\right\|+\left(c_{1}+\kappa_{1}+l_{21}-1\right)\left\|e_{2}\right\|\\ +\left\|e_{3}\right\|+\left(c_{2}+1\right)\mu \end{array}\right)\\ & \leq -k_{l}s^{2}-k_{s}\left|s\right|+s\left(\begin{array}{l} \left(c_{1}+c_{2}l_{11}-c_{2}+l_{12}\right){\hat{\lambda }}_{1}^{\max}+\left(c_{1}+\kappa_{1}+l_{21}-1\right){\hat{\lambda }}_{2}^{\max}\\ +{\hat{\lambda }}_{3}^{\max}+\left(c_{2}+1\right)\mu \end{array}\right)\\ & \leq -k_{l}s^{2}+\left(\begin{array}{l} \left(c_{1}+c_{2}l_{11}-c_{2}+l_{12}\right){\hat{\lambda }}_{1}^{\max}+\left(c_{1}+\kappa_{1}+l_{21}-1\right){\hat{\lambda }}_{2}^{\max}\\ +{\hat{\lambda }}_{3}^{\max}+\left(c_{2}+1\right)\mu -k_{s} \end{array}\right)\left|s\right|\\ & \leq -\left|s\right|\left(k_{l}\left|s\right|-\left(\begin{array}{l} \left(c_{1}+c_{2}l_{11}-c_{2}+l_{12}\right){\hat{\lambda }}_{1}^{\max}+\left(c_{1}+\kappa_{1}+l_{21}-1\right){\hat{\lambda }}_{2}^{\max}\\ +{\hat{\lambda }}_{3}^{\max}+\left(c_{2}+1\right)\mu -k_{s} \end{array}\right)\right) \end{array}$$

According to Equation (35), it is obvious that after a sufficiently long time |s| is bounded such that
(36)$$\begin{array}{l} \left\{ \begin{array}{l} \left|s\right|\leq {\hat{\lambda }}_{s}^{\max}\\ {\hat{\lambda }}_{s}^{\max}=A\left[\begin{array}{c} {\hat{\lambda }}_{1}^{\max}\\ {\hat{\lambda }}_{2}^{\max}\\ {\hat{\lambda }}_{3}^{\max}\\ 1 \end{array}\right] \end{array}\right.\\ where\;A=\left[\begin{array}{cccc} \frac{\left(c_{1}+c_{2}l_{11}-c_{2}+l_{12}\right)}{k_{l}} & 0 & 0 & 0\\ 0 & \frac{\left(c_{1}+\kappa_{1}+l_{21}-1\right)}{k_{l}} & 0 & 0\\ 0 & 0 & \frac{1}{k_{l}} & 0\\ 0 & 0 & 0 & \frac{\left(c_{2}+1\right)\mu -k_{s}}{k_{l}} \end{array}\right] \end{array}$$

Clearly, when increasing $k_{l}$ or adjusting the controller parameters and observer gains, |s| can converge to zero and $\dot{V}\left(s,\dot{s}\right)\leq 0$. Furthermore, when increasing $k_{l}$, the control input u in (23) does not increase greatly because that s is held close to 0 for all time t.

**Theorem** **2.**
*Based on the system (11), when the sliding mode surface is formulated as Equation (17), the IGC system is stable with controller (20), and EDO (21). The state will be converged to the following scopes.*


(37)$$\left\{ \begin{array}{l} \left|x_{1}\right|\leq \xi_{1}\\ \left|x_{2}+d_{1}\right|\leq \xi_{2} \end{array}\right.$$
where
(38)$$\left\{ \begin{array}{l} \xi_{1}=\left(2t_{12}\left(c2+1\right){\hat{\lambda }}_{1}^{\max}+2t_{12}{\hat{\lambda }}_{2}^{\max}+2t_{12}\hat{s}\left(0\right)e^{-\gamma t}\right)/q_{1}\\ \xi_{1}=\left(2t_{22}\left(c2+1\right){\hat{\lambda }}_{1}^{\max}+2t_{22}{\hat{\lambda }}_{2}^{\max}+2t_{22}\hat{s}\left(0\right)e^{-\gamma t}\right)/q_{2} \end{array}\right.$$

As long as $k_{l}$ is big enough, the sliding mode variable s will go to zero. Combined with Equation (17), when s=0, it can be written as follows.
(39)$$x_{3}=-c_{1}x_{1}-\left(c_{2}+\kappa_{1}\right)x_{2}-c_{2}{\hat{d}}_{1}^{*}-{\hat{\dot{d}}}_{1}^{*}-{\hat{d}}_{2}^{*}+\hat{s}\left(0\right)e^{-\gamma t}$$

Then, substituting Equation (39) for Equation (11), yields
(40)$$\left\{ \begin{array}{l} {\dot{x}}_{1}=x_{2}+d_{1}^{*}\\ {\dot{x}}_{2}=-c_{1}x_{1}-\kappa_{1}x_{2}-c_{2}{\hat{d}}_{1}^{*}-{\hat{\dot{d}}}_{1}^{*}+\hat{s}\left(0\right)e^{-\gamma t}+{\tilde{d}}_{2}^{*} \end{array}\right.$$

Now, defining a vector $x={\left[x_{1},x_{2}+d_{1}\right]}^{T}$ and a Lyapunov function is as follows.
(41)$$V_{x}=x^{T}Tx$$

Similarly, R is a positive define matrix which satisfies
(42)$$A^{T}T+TA=-Q$$

For any given positive define matrix Q, where A and T satisfies
(43)$$A=\left[\begin{array}{cc} 0 & 1\\ -c_{1} & -c_{2} \end{array}\right],T=\left[\begin{array}{cc} t_{11} & t_{12}\\ t_{11} & t_{22} \end{array}\right]$$

According to above equations, when $Q=diag\left(q_{1},q_{2}\right),\left(q_{1}>0,q_{2}>0\right)$, then T will satisfy the following condition.
(44)$$\left\{ \begin{array}{l} t_{11}=c_{1}t_{22}+c_{2}t_{12}\\ t_{12}=t_{21}=q_{1}/2c_{1}\\ t_{22}=\left(q_{2}+2t_{12}\right)/2c_{2} \end{array}\right.$$

Combining with Equation (40) and differentiating ${\dot{V}}_{x}$, then
(45)$$\begin{array}{ll} {\dot{V}}_{x} & =2t_{11}x_{1}{\dot{x}}_{1}+2t_{12}{\dot{x}}_{1}\left(x_{2}+d_{1}\right)+2t_{12}x_{1}\left({\dot{x}}_{2}+{\dot{d}}_{1}\right)+2t_{22}\left(x_{2}+d_{1}\right)\left({\dot{x}}_{2}+{\dot{d}}_{1}\right)\\ & =2t_{11}x_{1}\left(x_{2}+d_{1}\right)+2t_{12}{\left(x_{2}+d_{1}\right)}^{2}+2t_{12}x_{1}\left({\dot{x}}_{2}+{\dot{d}}_{1}\right)+2t_{22}\left(x_{2}+d_{1}\right)\left({\dot{x}}_{2}+{\dot{d}}_{1}\right) \end{array}$$

According to Equations (40) and (23), where
(46)$$\begin{array}{ll} {\dot{x}}_{2}+{\dot{d}}_{1} & =-c_{1}x_{1}-\kappa_{1}x_{2}-c_{2}{\hat{d}}_{1}^{*}-{\hat{\dot{d}}}_{1}^{*}+\hat{s}\left(0\right)e^{-\gamma t}+{\tilde{d}}_{2}^{*}+{\dot{d}}_{1}\\ & =-c_{1}x_{1}-\kappa_{1}x_{2}-c_{2}{\hat{d}}_{1}^{*}+\hat{s}\left(0\right)e^{-\gamma t}+{\tilde{d}}_{2}^{*}+{\tilde{\dot{d}}}_{1}^{*} \end{array}$$

According to Equation (23), then $x_{2}+{\hat{d}}_{1}^{*}=x_{2}+d_{1}^{*}-{\tilde{d}}_{1}^{*}$. Combining with Equation (44) and Equation (46), yields
(47)$$\begin{array}{l} {\dot{V}}_{x}=2\left(c_{1}t_{22}+c_{2}t_{12}\right)x_{1}\left(x_{2}+d_{1}\right)+2t_{12}{\left(x_{2}+d_{1}\right)}^{2}+\\ \;2t_{12}x_{1}\left(-c_{1}x_{1}-\kappa_{1}x_{2}-c_{2}{\hat{d}}_{1}^{*}+\hat{s}\left(0\right)e^{-\gamma t}+{\tilde{d}}_{2}^{*}+{\tilde{\dot{d}}}_{1}^{*}\right)+\\ \;2t_{22}\left(x_{2}+d_{1}\right)\left({\dot{x}}_{2}-c_{1}x_{1}-\kappa_{1}x_{2}-c_{2}{\hat{d}}_{1}^{*}+\hat{s}\left(0\right)e^{-\gamma t}+{\tilde{d}}_{2}^{*}+{\tilde{\dot{d}}}_{1}^{*}\right)\\ \;=-q_{1}x_{1}^{2}+2t_{12}x_{1}\left(x_{2}{\tilde{d}}_{1}^{*}+{\tilde{\dot{d}}}_{1}^{*}+{\tilde{d}}_{2}^{*}+\hat{s}\left(0\right)e^{-\gamma t}\right)-q_{2}{\left(x_{2}+d_{1}\right)}^{2}+\\ \;2t_{22}\left(x_{2}+d_{1}\right)\left(c_{2}{\tilde{d}}_{1}^{*}+{\tilde{\dot{d}}}_{1}^{*}+{\tilde{d}}_{2}^{*}+\hat{s}\left(0\right)e^{-\gamma t}\right)\\ \;\leq -\left|x_{1}\right|\left(q_{1}\left|x_{1}\right|-\chi_{11}\right)-\left|x_{2}+d_{1}\right|\left(q_{2}\left|x_{2}+d_{1}\right|-\chi_{12}\right) \end{array}$$
where
(48)$$\left\{ \begin{array}{l} \chi_{11}\leq 2t_{12}\left(c2+1\right){\hat{\lambda }}_{1}^{\max}+2t_{12}{\hat{\lambda }}_{2}^{\max}+2t_{12}\hat{s}\left(0\right)e^{-\gamma t}\\ \chi_{12}\leq 2t_{22}\left(c2+1\right){\hat{\lambda }}_{1}^{\max}+2t_{22}{\hat{\lambda }}_{2}^{\max}+2t_{22}\hat{s}\left(0\right)e^{-\gamma t} \end{array}\right.$$

When the states are satisfied as
(49)$$\left\{ \begin{array}{l} \left|x_{1}\right|>\xi_{1}\\ \left|x_{2}+d_{1}\right|>\xi_{2} \end{array}\right.$$

The following can be obtained
(50)$${\dot{V}}_{x}\leq 0$$

Thus, according to Lyapunov stability theory, the IGC system is stable, and the convergence scope of the states in sliding mode is obtained as follows.
(51)$$\left\{ \begin{array}{l} \left|x_{1}\right|\leq \xi_{1}\\ \left|x_{2}+d_{1}\right|\leq \xi_{2} \end{array}\right.$$

## 4. Simulation

To verify the effectiveness and interception performance of the proposed method under different scenarios, several simulations will be discussed in this section. The initial values of the missile are as follows.
(52)$$\begin{array}{l} \kappa_{1}=0.3487,\kappa_{2}=-0.2741,\kappa_{3}=-17.80\\ \kappa_{5}=0.068,b=-31.267 \end{array}$$

In addition, the initial relative distance between the missile and the target is $R_{0}=10\;\mathrm{Km}$. The constant velocity of the missile and target is 500 m/s and 250 m/s, respectively. The initial flight path angle for missile and target is $\theta_{m0}=45^{\circ }$ and $\theta_{t0}=120^{\circ }$ respectively. The control constraint is set as $\left|\delta_{z}\right|\leq 15^{\circ }$. The initial value of the states is ${\left[x_{1}\left(0\right),x_{2}\left(0\right),x_{3}\left(0\right)\right]}^{T}={\left[0,0,0\right]}^{T}$, and the simulation step size is 0.001 s. 

### 4.1. Case I

In this case, we will consider that the target maneuver is assumed to be $a_{T}=50\;*\;\cos(0.5t)$, which is shown in Figure 3. To further illustrate the effectiveness of the proposed method, this paper uses BS-SMC (Back-Stepping Sliding Mode Control) as a comparison. The simulation results are shown in Figure 4 and Figure 5.

As shown in the above figures, Figure 3 shows the maneuver characteristics of the target. Figure 4 shows the trajectory of the missiles and the target by using BS-SMC (Back-Stepping Sliding Mode Control) and SMC-EDO proposed in this paper. In addition, the LOS angular rate, the rudder deflection, the change of sliding mode surface, the pitch angular rate, and the pitch angle are shown in Figure 5a–e; Figure 5f shows the variation of the attack angle.

According to Figure 3 and Figure 4, despite existing the maneuver in velocity of the target, both methods can intercept the target. However, it is obvious that compared with the BS-SMC method, the method proposed in this paper can intercept the target earlier in the same conditions. Using the SMC-EDO method proposed in this paper, the interception performance is better, and the Missile—Target relative distance is 0.3571 m and 1.398 m respectively.

Figure 5a shows the change of the LOS angular rate by using the BS-SMC (Back-Stepping Sliding Mode Control) and the SMC-EDO proposed in this paper. We can conclude that the LOS angular rate can be soon converged to zero. However, by exhaustive analysis of the figure, it is found that the convergence time of the LOS angular rate for the two methods is different; the proposed IGC method is the faster. The reason is that there is a time delay in the SMC-BS method, while the IGC method proposed in this paper has no time delay. This verifies Remark 1, on the other hand. In addition, during the whole interception process, the fluctuation is smaller especially when approaching the target.

Figure 5b shows the variation of rudder deflection by using these two methods. It is clear that the rudder deflection angle is smaller in the whole process, compared with the SCM-BS method. In addition, the amplitude of the rudder is relatively smaller and smoother, and the rudder response is faster.

Figure 5c shows the change of the sliding mode surface and it is clear that the sliding surface can converge close to 0 quickly, compared with another method. Moreover, the reason why the surface appears to have sinusoidal fluctuation is that the target is in the sinusoidal maneuver.

Figure 5d provides the change of the pitch angular rate. Compared with the SCM-BS method, the amplitude of the pitch angular rate is relatively smaller and smoother.

Combining Figure 5e and Figure 5f clearly shows that the missile is very stable during the whole interception process and the change of the pitch angle is very smooth. It also shows that the control performance is better, when using the proposed IGC method in this paper.

### 4.2. Case II

In order to further verify the feasibility of this method, the missile will intercept the strong maneuvering target. In this case, it will increase the difficulty of the target’s velocity assumed to be $v_{T}=400+50\sin\left(0.5t\right)$ and the trajectory inclination angle is assumed to be $\theta_{T}{=120}^{\circ }{+60}^{\circ }\sin\left(0.15t\right)$, which is shown in Figure 6. Additionally, the additive external disturbances $d_{2}^{*}=0.5\sin\left(t\right)$$d_{3}^{*}=0.2\sin\left(t\right)$ are also added. Similarly, the SMC-BS method is also as a comparison in this case. The simulation results are presented in Figure 7.

The above figures show the trajectory of the missiles and target, the LOS angular rate, the rudder deflection, the change of sliding mode surface, and the pitch angular rate (shown in Figure 7a–e), and Figure 7f shows the variation of the attack angle.

According to Figure 6, it is obvious that the maneuverability of the target is greater than *CASE I*. Combining with Figure 7a, it is obvious that even with increasing the interception difficulty, the missile can also intercept the maneuvering target by using these two methods. To be more specific, the method proposed in this paper can also intercept the target earlier and the interception performance is better, and the Missile—Target relative distance is 0.531 m and 1.5290 m respectively.

Figure 7b shows the change of the LOS angular rate by using the BS-SMC method and the SMC-EDO method proposed in this paper. It is obvious that the LOS angular rate can be soon converged to zero. However, by exhaustive analysis of the figure, it is clear that the convergence time of the LOS angular rate for the two methods is different; the SMC-EDO method is the faster. The reason is that there is time delay in the SMC-BS method, while the IGC method proposed in this paper has no time delay. In addition, during the whole interception process, the fluctuation is smaller especially when approaching the target.

Figure 7c provides the variation of rudder deflection by using the BS-SMC method and the SMC-EDO method proposed in this paper. It is clear that the rudder deflection angle is smaller in the whole process, compared with the SCM-BS method. The amplitude of the rudder is relatively smaller and smoother, especially when approaching the target. In addition, we can also see that the rudder response is faster.

Figure 7d shows the change of the sliding mode surface by using these two methods, and it is clear that the sliding surface can also converge to near 0 quickly. The fluctuation is also smaller, which illustrates that the missile is relatively stable during the whole interception process.

The change of the pitch angular rate is shown in Figure 7e. Compared with the SCM-BS method, the amplitude of the pitch angular rate is also relatively smaller and smoother, especially when approaching the target.

Figure 7f shows the variation of the attack angle, and it is clear that the change of the attack angle is very smooth. It also shows the control performance is better, when using the proposed IGC method in this paper.

Therefore, the use of the proposed IGC method, which achieves the real concept of IGC, shortens the response time and has a high interception accuracy.

### 4.3. Case III (Monte Carlo Simulations)

To further demonstrate the robustness of the proposed IGC control law, the Monte Carlo simulation is conducted with 500 sampling points. The lumped disturbances $d_{3}^{*}=0.5\sin\left(t\right)$ and $d_{4}^{*}=0.2\sin\left(t\right)$ were added to the IGC model. The additional conditions are imposed as per Table 1. In this case, the SMC-BS method is also selected as a contrast.

The simulation results are presented in Figure 8, and the interception performance of these two methods is shown in Table 2.

The above figures provide the miss distance of the missile and the target by using the MSC-BS method and the SMC-EDO method. Using these two methods, the interception performance is shown in Table 2.

As can be seen in Figure 8a, it is clear that the missile impacting points are relatively concentrated by using the SMC-EDO method proposed in this paper. By extension, it is obvious that the most impacting points of the SMC-EDO method are smaller than the SMC-BS method and the impacting radius is also smaller.

Figure 8b provides the M—T relative distance for each time. It is obvious that all of these impacting points are less than 0.5 m, using the proposed IGC control law in this paper. This also reflects that the method has a higher interception accuracy from the side.

Table 2 shows the interception performance by using these two methods. Both the Mean error and the MSE of M—T distance are smaller by using the proposed IGC control method in this paper. In addition, the missile intercepting time is also smaller, which illustrates that the missile can intercept the target earlier under the same conditions.

## 5. Conclusions

To solve the problem of intercepting the great maneuvering target and reducing the interception response time, a new control algorithm for an integrated guidance and control (IGC) system is proposed in this paper by combining the sliding mode and the extended disturbance observer. After formulating the Missile—Target problem, the paper establishes the uncertain IGC system. In addition, to intercept the maneuvering target, the nonlinearities, the perturbations, and the maneuvering of the target are regarded as disturbance. To estimate the disturbance and their derivatives, the paper designs a second-order disturbance observer. This, combined with the second-order disturbance observer and the sliding mode control method, a new sliding mode surface is designed to obtain the rudder deflection command directly to achieve the real concept of the IGC system. Then, the paper proves the stability of the system. Finally, several simulation cases are provided to demonstrate the superiority of the proposed method in reducing the response time, increasing the rudder response, decreasing the intercepting time, and improving the interception accuracy.

## Figures and Tables

**Figure 1 sensors-22-07618-f001:**
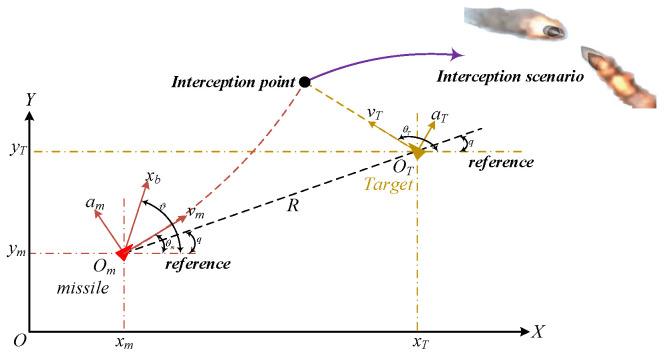
The relative motion of the missile and target.

**Figure 2 sensors-22-07618-f002:**
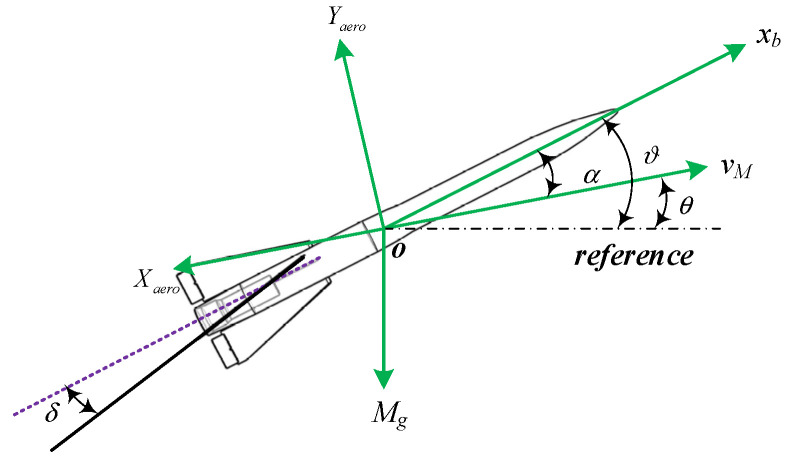
Dynamics of the missile in the pitch plane.

**Figure 3 sensors-22-07618-f003:**
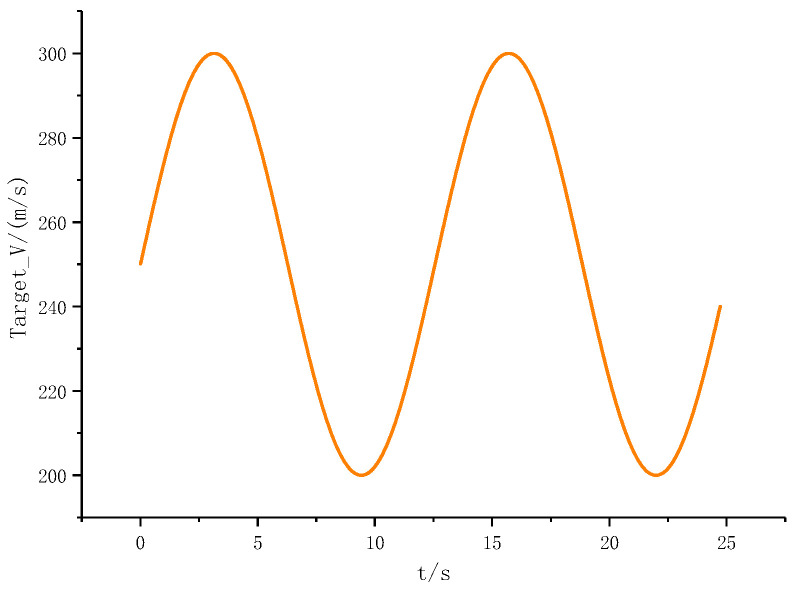
The maneuver characteristics of the target.

**Figure 4 sensors-22-07618-f004:**
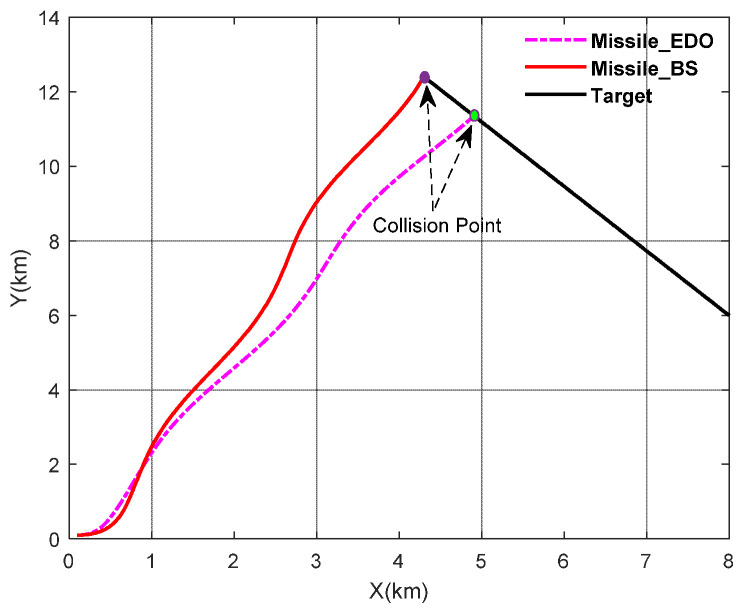
Missile–Target pursuit trajectory by using EDO and SMCBS.

**Figure 5 sensors-22-07618-f005:**
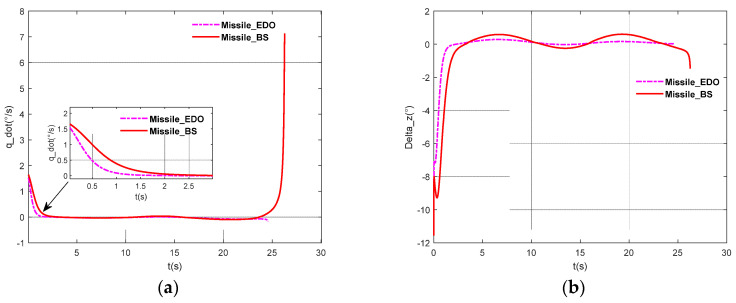
(**a**) The curve of LOS angular rate; (**b**) The curve of rudder deflection; (**c**) The curve of the sliding mode surface proposed in this paper; (**d**) The curve of the pitch angular rate; (**e**) The curve of the pitch angle; (**f**) The curve of the attack angle.

**Figure 6 sensors-22-07618-f006:**
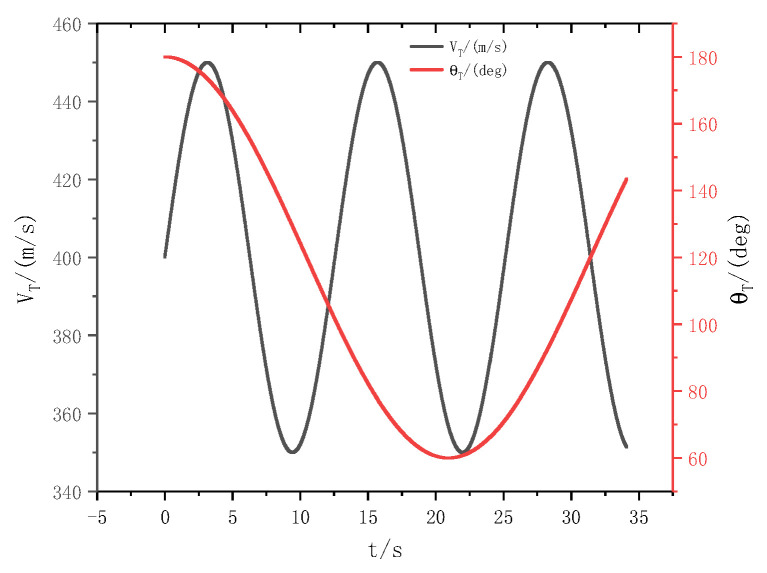
The maneuver characteristics of the target.

**Figure 7 sensors-22-07618-f007:**
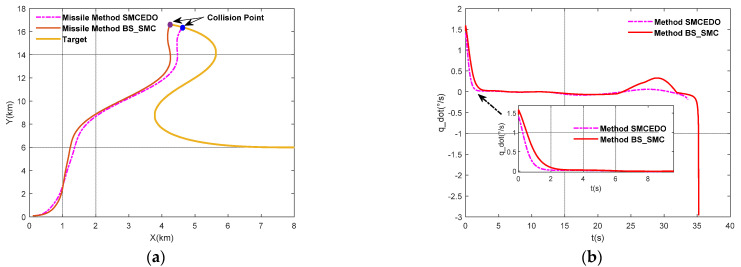
(**a**) Missile–Target pursuit trajectory; (**b**) The curve of the LOS angular rate; (**c**) The curve of the rudder deflection; (**d**) The curve of the sliding mode surface; (**e**) The curve of the pitch angular rate; (**f**) The curve of the attack angle.

**Figure 8 sensors-22-07618-f008:**
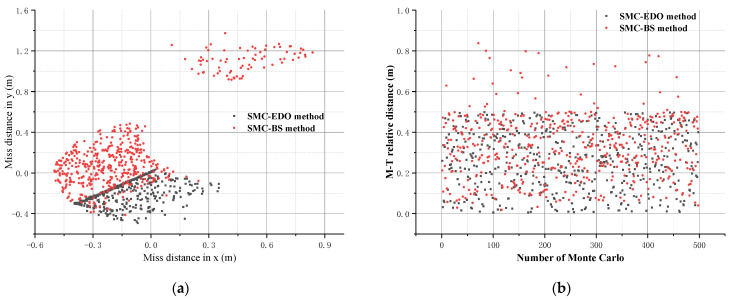
(**a**) The miss distance in x and y; (**b**) The characteristic of the relative distance of M—T.

**Table 1 sensors-22-07618-t001:** The Monte Carlo parameters under case III.

Symbol	Quantity	Values
$\theta_{m}$	flight path angle of M	40∼50(deg)
$v_{m}$	velocity of the M	470∼530 (m/s)
$\omega_{z}$	the rate of pitch angle	−3∼3(deg/s)
$x_{m}$	Initial position in *x*	0∼100(m)
$y_{m}$	Initial position in *y*	0∼100(m)
$\vartheta_{m}$	pitch angle	−3∼3(deg)
$x_{T}$	Initial position in *x*	7500∼8500(m)
$y_{T}$	Initial position in *y*	5500∼6500(m)
$v_{T}$	velocity of the T	350~450+50sin(0.5t)(m/s)
$\theta_{T}$	flight path angle of T	$115^{\circ }∼125^{\circ }{+60}^{\circ }\sin\left(0.15t\right)\left(\deg\right)$
$\delta_{z}$	rudder deflection	−3∼3(deg)

**Table 2 sensors-22-07618-t002:** The interception performance under case IV.

Method	SMC-EDO	SMC-BS
Mean error of M-T distance	0.2643 (m)	0.4901 (m)
MSE of M-T distance	0.1430 (m)	0.2139 (m)
Intercepting time	16.35 s	19.28 s

## Data Availability

Not applicable.

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
