# Peer review of "A New Sliding Mode Control Algorithm of IGC System for Intercepting Great Maneuvering Target Based on EDO"

_sensors, 2022, doi:10.3390/s22197618_

Round 1

Reviewer 1 Report

The control method for IGC system is important for missile applications. This manuscript may be suitable for publication after the following issues are addressed.
(1) The authors' introduction to IGC concepts and functions is clear, but the introduction to the IGC field literature is confusing and lacks mainline logic.

(2) What is the full name of BC? Does it mean the same "BC" in "RABC"?

(3) How constraints on control inputs are considered?

(4) The authors present the modified sliding surface as their new contribution. However, it is not stated what the traditional sliding surface is used as comparison base.

(5) In fact, the sliding mode method is widely used in missile control, and the authors need to better summarize their new contributions to the design of sliding mode surfaces.

Ref:

Yu H, Dai K, Li H, et al. Three-dimensional adaptive fixed-time cooperative guidance law with impact time and angle constraints[J]. Aerospace Science and Technology, 2022, 123: 107450.
(6) Lack of theoretical analysis of convergence rate.

(7) Lack of performance comparison with literature in simulation.

(8) Can the proposed method solve all types of high-speed intersection problems? What determines the upper limit of its performance?

Author Response

Firstly, I give my heartfelt thanks to all reviewers for reviewing the manuscript and offering interesting suggestions. We have studied all reports carefully. For each suggestion, we revised the manuscript carefully in in the attachment.

I wonder if these modifications are to your satisfaction. If you have any questions, please contact me at any time. Thank you very much. 

best wishes to you.

Reviewer 2 Report

The paper introduces a new control algorithm for integrated guidance and control systems that combines a sliding mode control synthesis with a disturbance state estimator. 

Overall the manuscript reads like a laboratory report of a rather flaky simulation of a closed loop system. It is rather hard to follow as many details are not mentioned and there is no motivation in choosing the design elements involved (i.e. choice of the sliding surface, sliding conditions fulfilment, state-estimations/observer design). Although there is a limited analysis of the results in the three cases/simulations, I didn't see a clear reduction in response time, rudder response increase, or a motivated interception probability, except for the superficial analysis in Figure 5.

More particular observations:

- the English language is rather hard to follow and the formulations have typos, I would recommend an English proofing service before submission to avoid typos, colloquial language and ambiguous formulations (e.g. "many literatures", "there are lots of ways", "a new modified...surface", "mismatched uncertain", "the sliding surface will be soon converged to zero before 2.5s" )

- there are 2.5 pages on state-of-the-art w.r.t the problem formulation and how the introduction relates to the actual problem to solve and the innovative aspects of the proposed solution w.r.t state-of-the-art. Very important, there is no comparison against other controllers, although the conclusion states "superiority of the method".

- in formula 11 the d_star terms look like simple regularizers which are computed with many "magic numbers" which are neither motivated nor explained

- in equation 11 when introducing the core element of the sliding mode controller, there is no motivation and explanation of the choice of sliding surface: basically the place and the dynamics where the control law should push the system state trajectory - how do you choose the shape of the surface, how about the gain margin, where is the sliding condition and convergence condition?

- In equation 20 When is saturation becoming active?? In the vicinity of the sliding surface (the invariant dynamics), there is a space where the system trajectory reaches and will go under the impact of the sign (discontinuity) function action. The sat function should be acting differently w.r.t. the position of the sliding surface.

- I would have also added the asymptotic stability analysis using the Lasalle principle, to study the closed-loop stability

- the simulations are not relevant, as they only depict very controlled experiments. There are some points, which make me wonder: for instance the bump in Figure 3 d at the time [0, 2.5] and in Figure 3 f and h matching the bumps in alpha and omega; there is a formalized way to write units (X(km)) in SI; separate Figure 3 in two figures;

- the authors claim "why the surface appears sinusoidal fluctuation is that the target is in sinusoidal manoeuvre"- when the surface by definition uses the error vector (reference vector - state vector) it is typically not related to the shape of the sliding surface - as this is only a function of the difference and not only a function of the reference.

Typos:

- line 160: planer - planner

- line 173, 177 missiel - missile

- line 320 addictive - additive

Finally, I am not able to endorse the publication of this manuscript as I cannot grasp the novelty, superiority, and soundness of the proposed approach.

Author Response

Dear Editors and Reviewers

Firstly, I give my heartfelt thanks to all reviewers for reviewing the manuscript and offering interesting suggestions. We have studied all reports carefully. For each suggestion, we revised the manuscript carefully. All revisions are listed In the attachment .

I wonder if all these responses are to your satisfaction. If you have any questions, please contact me at any time. Thank you very much.

best wishes to you.

Round 2

Reviewer 1 Report

The authors have well addressed all my concerns. The manuscript is suitable for publication now.

Reviewer 2 Report

The authors addressed all my concerns and put a lot of effort into rewriting the manuscript. Now it is in better shape.